# From Evidence to Design Solution—On How to Handle Evidence in the Design Process of Sustainable, Accessible and Health-Promoting Landscapes

**Marie C. Gramkow** [1,*] **, Ulrik Sidenius** [1] **, Gaochao Zhang** [2,3] **and Ulrika K. Stigsdotter** [1]

1    Section for Landscape Architecture and Planning, Department of Geosciences and Natural Resource Management (IGN), Faculty of Science, University of Copenhagen, Rolighedsvej 23, 1958 Frederiksberg C, Denmark; us@ign.ku.dk (U.S.); uks@ign.ku.dk (U.K.S.)
2    Ministry of Education Key Laboratory for Earth System Modeling, Department of Earth System Science, Tsinghua University, Beijing 100084, China; lainong@foxmail.com
3    Center for Healthy Cities, Institute for China Sustainable Urbanization, Tsinghua University, Beijing 100085, China
*    Correspondence: mcg@ign.ku.dk

**Abstract:** The work of landscape architects can contribute to the United Nation's Sustainable Development Goals and the associated 'Leave no one behind' agenda by creating accessible and health-promoting green spaces (especially goals 3, 10 and 11). To ensure that the design of green space delivers accessibility and intended health outcomes, an evidence-based design process is recommended. This is a challenge, since many landscape architects are not trained in evidence-based design, and leading scholars have called for methods that can help landscape architects work in an evidence-based manner. This paper examines the implementation of a process model for evidence-based health design in landscape architecture. The model comprises four steps: 'evidence collection', 'programming', 'designing', and 'evaluation'. The paper aims to demonstrate how the programming step can be implemented in the design of a health-promoting nature trail that is to offer people with mobility disabilities improved mental, physical and social health. We demonstrate how the programming step systematizes evidence into design criteria (evidence-based goals) and design solutions (how the design criteria are to be solved in the design). The results of the study are presented as a design 'Program', which we hope can serve as an example for landscape architects of how evidence can be translated into design.

**Keywords:** accessibility; design process; design research; evidence-based health design; green spaces; human health; landscape architecture; leave no one behind; user-centred design

## 1. Introduction

### 1.1. Landscape Architecture and Sustainable Development

The work of architects, landscape architects, planners, and designers can contribute to achieving the United Nations' (UN) 17 Sustainable Development Goals (SDGs). Some of the SDGs are easier to achieve by using architecture than others. For example, SDG 11, 'Sustainable cities and communities', relates to challenges connected to rapid urbanization. Making cities sustainable relates to the work of architects, landscape architects, and city planners since it includes 'safe and affordable housing', 'creating green public spaces', and 'improving urban planning and management' [1].

The World Health Organization (WHO) has identified target 7 in SDG 11, "By 2030, provide universal access to safe, inclusive and accessible, green and public spaces, in particular for women and children, older persons and persons with disabilities" [2], as a framework that can help to promote the agenda on the relationship between urban green space and human health.

Increasing numbers of studies indicate the positive health impact of green spaces on various groups of people, for example elderly people [3], teenagers [4], war veterans [5], immigrants [6], and people with eating disorders [7]. However, one group of people has largely been overlooked in the research field of nature and human health relationship [8]—people with disabilities. People with disabilities represent approximately 15 percent of the world's population [9]. That corresponds to over a billion people [9]. Disability is diverse, but it is calculated that between 110 to 190 million people 15 years and older have substantial problems in functioning [9].

### 1.2. Landscape Architecture and 'Leave No One behind'

In the 17 SDGs, people with disabilities are mentioned in five of the goals (SDG 4, 8, 10, 11, and 17). In relation to the Agenda for Sustainable Development and its SDGs, a promise has been formulated: Leave no one behind. 'Leave no one behind' represents the commitment across the SDGs to eliminate discrimination and exclusion and reduce inequalities that leave people behind [10]. A recent study shows that people with mobility disabilities visit green space less frequently than the able-bodied population in Denmark [11]. The study further shows that there is a relation between the frequency of visits to green space and the health-related quality of life for both the able-bodied population and people with mobility disabilities [11]. Mobility disabilities are the most common type of disabilities in Denmark [12], and people with motor disabilities have significantly poorer health-related quality of life compared to the able-bodied population [12]. Mobility disabilities may include impairments in the limbs, back, neck, hands, or feet and can be congenital or acquired later in life. Mobility disabilities vary both in severity and in the extent to which people with them require assistive devices. One might say that people with mobility disabilities are left behind in relation to target 7 of SDG 11 'Universal access to safe, inclusive and accessible, green and public spaces', but also in relation to SDG 3 'Good health and well-being' and SDG 10 'Reduced inequalities'. A recent report on SDG progress states that persons with disabilities "continue to face multiple disadvantages, denying them both life opportunities and fundamental human rights" [13]. The WHO also states that people with disability are seldom targeted in health promotion and prevention activities [9].

The UN Convention on the Rights of Persons with Disabilities (CRPD) states that people with disabilities, including those with various impairments which due to different barriers, are hindered in participating in society on the same level as others [14]. Disability thus appears when the physical surroundings do not meet the needs of the individual; it could appear when one is not able to enter a building due to high steps or when one cannot cross a stream in a park due to a bridge that is too steep. The design of the environment can therefore affect whether a person with a mobility impairment experiences themselves being 'handicapped' (Figure 1). This leaves architects, landscape architects, and city planners with a great responsibility.

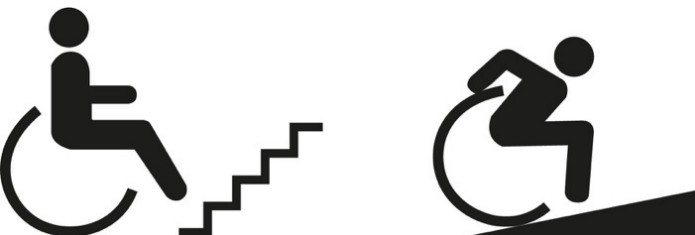

**Figure 1.** The environment makes the person to the left 'handicapped', while the person to the right is not.

### 1.3. Landscape Architecture and Human Health

Landscape architects must create accessible designs of green space for people with disabilities to avoid them being 'handicapped' and 'left behind'. Furthermore, green space

that is accessible to people with disabilities can promote their physical, mental, and social health [8].

Health design in landscape architecture is defined by Stigsdotter [15] as the deliberate design of green spaces so that the design, in a predetermined way, supports health processes and results in improved health outcomes. This definition has a salutogenic (health-creating) standpoint where the design's emphasis is to support the processes of overcoming illness by focusing on the users' capabilities, capacities, and strengths—irrespective of diagnosis [15]. However, when designing for users with diagnoses, it is fundamental for the landscape architect to understand that sick and vulnerable persons may perceive and interpret the environment differently from healthy users [16]. If the landscape architect does not have knowledge of the user groups' characteristics, the design may not support their health, and it can actually worsen the users' health condition [17]. Also, how the users will use the green space must be carefully considered in the design. If the users do not use the green space as planned, the intended positive health outcomes will be reduced [15]. According to the WHO, it is important that landscape architects understand that green space should be designed in a manner that supports and improves human health [18], as defined by the WHO [19]. To attain successful health design in landscape architecture, Stigsdotter and Sidenius propose that designers incorporate four foundation stones in the design process: 1. the target group, 2. the nature and human health relationship, 3. the environment, and 4. how the target group will use the environment [20]. They further recommend that an evidence-based design process is followed in order to ensure the design will support health, and describe a comprehensive evidence-based approach to health design in landscape architecture, the Evidence-Based Health Design in Landscape Architecture (EBHDL) process model [20]. Some of the identified benefits of the EBHDL process model are that it is user-centered (the users are central to the design process), interdisciplinary (incorporates evidence from several research and practice disciplines), systematic (the evidence is systematized in relation to clear aims), and transparent (it provides documentation of the arguments governing the decisions made regarding the design) [20].

*1.4. Evidence-Based Landscape Architecture*

Landscape architecture has increasingly become a more evidence-based profession [21], and by working in an evidence-based manner, we may ensure compatibility between the design and the users' characteristics. Some parts of landscape architecture have a tradition of working in an evidence-based manner, for example in landscape technology that works with storm water management [22]. Working in an evidence-based manner in design processes derives from other disciplines where decisions are guided by research. Most significant in this case is evidence-based medicine, which integrates best available research evidence with clinical expertise [23]. The necessity of working in an evidence-based manner in health care architecture became apparent in the early 2000s when two reports on hospital deaths and hospital acquired illnesses were published [24]. Stichler and Hamilton [25] (p. 3) initially formalized the concept of evidence-based design and defined it as: "[ . . . ] a process for the conscientious, explicit, and judicious use of current best evidence from research and practice in making critical decisions, together with an informed client, about the design of each individual and unique project." In 2011 Brown and Corry [21] (p. 328) defined Evidence-Based Landscape Architecture (EBLA) as "the deliberate and explicit use of scholarly evidence in making decisions about the use and shaping of land". While Brown and Corry talk about scholarly evidence, Stichler and Hamilton's definition includes evidence that may not just be found in scholarly works, but also from practice. Such evidence from practice may help address the gaps in the literature on the relatively new field of EBLA [26], which are related to the inconsistent quality of research in landscape architecture [27]. In relation to evidence-based health design in landscape architecture, there is also a lack of relevant and applicable research evidence on nature and human health relationship that can be used by practicing landscape architects [15].

According to Stichler and Hamilton [25], the evidence-based design process varies between landscape architectural companies and it remains unclear how the evidence is acquired, assessed, and used. Several researchers and practitioners have developed approaches to working with an evidence-based approach [27]. Some have broadened the evidence concept to include experiences and best practices [28,29]. When working in an evidence-based manner, the landscape architect may lack knowledge and skills about how to identify and handle evidence [22], which can be especially challenging when the evidence stems from various sources, for example, design guidelines, research results, and theories. Furthermore, the landscape architect may lack knowledge about how to integrate the evidence into a design. Therefore Brown and Corry [22] (p. 9) recommend future research to "[ . . . ] provide specific steps or guiding questions to assist landscape architects in using EBLA."

### 1.5. Research Goals and Contribution to the Field

This paper examines the evidence-based approach to health design in landscape architecture, the EBHDL process model, developed during the last 15 years by an interdisciplinary research group at the University of Copenhagen [20]. The EBHDL model includes the whole design process from collecting evidence to post-occupancy evaluation [20].

This paper aims, for the first time, to demonstrate in detail how the 'Programming' step of the EBHDL process model can be conducted by applying it in the project 'the Move Green Nature Trail' [30]—an (to be established in 2021) health-promoting nature trail in Denmark that is to be accessible for people with mobility disabilities. To the authors' knowledge, this is the first time the EBHDL approach to designing a nature trail has been demonstrated. The trail is to be accessible for people with mobility disabilities. The paper contributes to the field of landscape architecture and sustainability by exemplifying how evidence (derived from various sources, e.g., peer-reviewed articles to landscape analyses) can be translated into health-promoting and accessible design solutions for people with mobility disabilities, and thereby contribute to the SDGs and the 'leave no one behind' agenda.

The current paper addresses the gap in the research literature as regards studies and interventions designed to assess and attain a positive health impact of green spaces on people with disabilities [8]. It does so by exploring how to design accessible and health-promoting green space that caters for people with mobility disabilities.

## 2. Method

### 2.1. The EBHDL Process Model

The EBHDL process model has been validated in various research projects and has been applied in the design of the University of Copenhagen's Nature, Health & Design Laboratory, which is a full-scale outdoor laboratory currently consisting of two projects: Nacadia® therapy garden and Octovia® health forest [31,32]. The EBHDL process model (Figure 2) includes four steps: 1. Evidence collection, 2. Programming, 3. Design, and 4. Evaluation.

The EBHDL process model can be applied to the design of any health-based design project in landscape architecture. The design process starts with collecting evidence related to four main topics that are directly linked to the four foundation stones of health design in landscape architecture [20]. 'Target group' evidence provides information on who the design is intended for. Examples of relevant evidence include gender, age, health status, health-related challenges, possible limitations, or strengths. 'Nature and human health relationship' evidence focuses on how the target group can benefit from exposure to nature, that is, the focus is on possible positive associations between nature exposure and human health. 'Environment' evidence focuses on the existing conditions on a given site and how the design can promote health. This draws on the landscape architect's practical and aesthetic skills and knowledge. 'Use of nature' evidence focuses on how the site can be

used to support the target group's health. This evidence helps illuminate how the target group can use nature as a form of treatment, for health promotion or to prevent sickness.

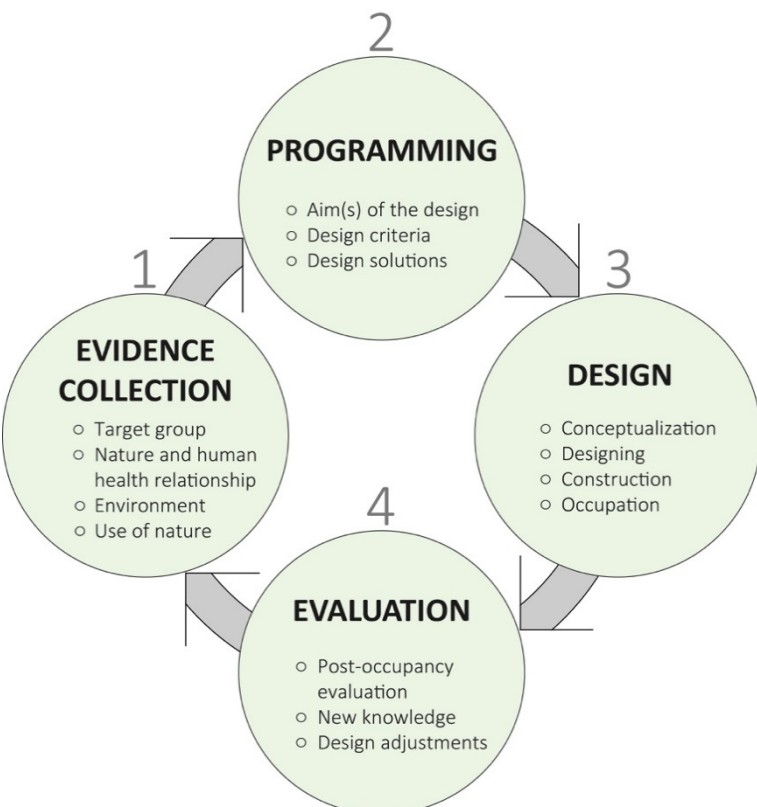

**Figure 2.** Schematic presentation of the Evidence-Based Health Design in Landscape Architecture (EBHDL) process model.

The broad collection of evidence in the EBHDL process model comes from a variety of sources [20]. Evidence in relation to 'target group' and 'nature and human health relationship' can be found in articles published in peer-reviewed scientific journals (so-called white literature), theoretical works from environmental psychology, landscape architecture, and psychology, in reports and working papers (so-called gray literature). Evidence can also be gathered by interviewing the target group, relatives of the target group, and experts. In relation to 'environment', evidence can also be gathered from landscape analyses, best cases, user and expert involvement, and regulations and legislation. Concerning 'use of nature', evidence can be found in 'white' and 'gray' literature, but also from best practices.

After the evidence has been collected, it is then systemized in the second step, 'programming', which consists of three elements: aim(s) of the design; design criteria; and design solution. The EBHDL process model presents the three programming elements and the systematized evidence from step one in a three-column table—the 'program' (for an example see Table A1). The first part of the programming 'Aim(s) of the design' identifies which health outcomes the design is to have an impact on. At this point, the EBHDL process requires that the possible therapeutic approach is clarified, for example, the type of treatment or health-promoting interventions. The heading of the first column of the EBHDL program table is 'Evidence', which sorts the evidence under the sub-headings from the first step of the EBHDL process model. The next column in the EBHDL program table is 'Design criteria', which translates the evidence into design criteria. These criteria are explicit goals based on the evidence. The final column in the EBHDL program table presents the 'Design solution' that describes how the design criteria are to be met.

The goal of the 'programming' is to guide the design process into the third step of the process model, the 'Design'. The design step comprises four parts: a. 'Conceptualizing', which refers to broad outlines (schematic drawings or descriptions) of function, structure, and form of the design criteria; b. 'Design'; c. 'Construction'; and d. 'Occupation', which is when the target group starts to use the site.

The fourth step of the EBHDL process model is 'Evaluation', which consists of three parts: a. 'Post-Occupancy Evaluation' (POE), which examines whether the design meets the original aim(s) set in 'programming'; b. 'New knowledge', which refers to the results and experiences from the POE that can be applied as new evidence in other projects; and c. 'Design adjustments', which refers to the new knowledge generated from the POE that can be used as a basis for making decisions on adjusting the design. At this point, the process returns to the first step 'Evidence Collection' in the EBHDL process model.

### 2.2. The Move Green Project

The Move Green research project explores which factors prevent people with mobility disabilities from using nature. The authors of the present paper are attached to the 'Move Green Research Project' at the University of Copenhagen's Department of Geosciences and Natural Resource Management. In the present study, we report on the initial design stages of the Move Green project's application of the EBHDL process model to the design and construction of a full-scale, health-promoting outdoor setting—the- 'Move Green Nature Trail'. The trail, which has not yet been constructed, is to be accessible for people with mobility disabilities and is to offer improved mental, physical, and social health through mental restoration, physical exercise, and social activities for all ages and genders within the target group. It is also intended to be used as a demonstration site and learning environment for landscape architects, planners, healthcare personnel (e.g., physiotherapists), and other relevant professions that work with accessibility, health promotion, and rehabilitation.

When completed, the Move Green Nature Trail will be located in the Hoersholm arboretum (Figure 3). Large parts of the area have served as a wildlife reserve, and today the vegetation is free-growing and wild. Different types of vegetation can be found (e.g., beech forest, mixed forest, marsh area) and the area also includes two large lakes that are rich in wildlife (Figure 4). The southwestern area is forested and has large elevation differences in the terrain. While some trails already exist, they often become waterlogged during the autumn, winter, and early spring (Figure 4).

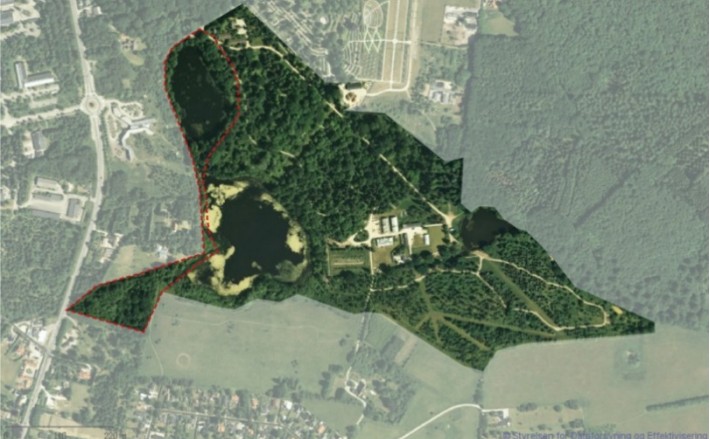

**Figure 3.** The map shows the Hoersholm Arboretum. The dark red line illustrates the area for the Move Green Nature Trail. (Contains data from the Danish Agency for Data Supply and Efficiency, Historical map 2008, December 2020).

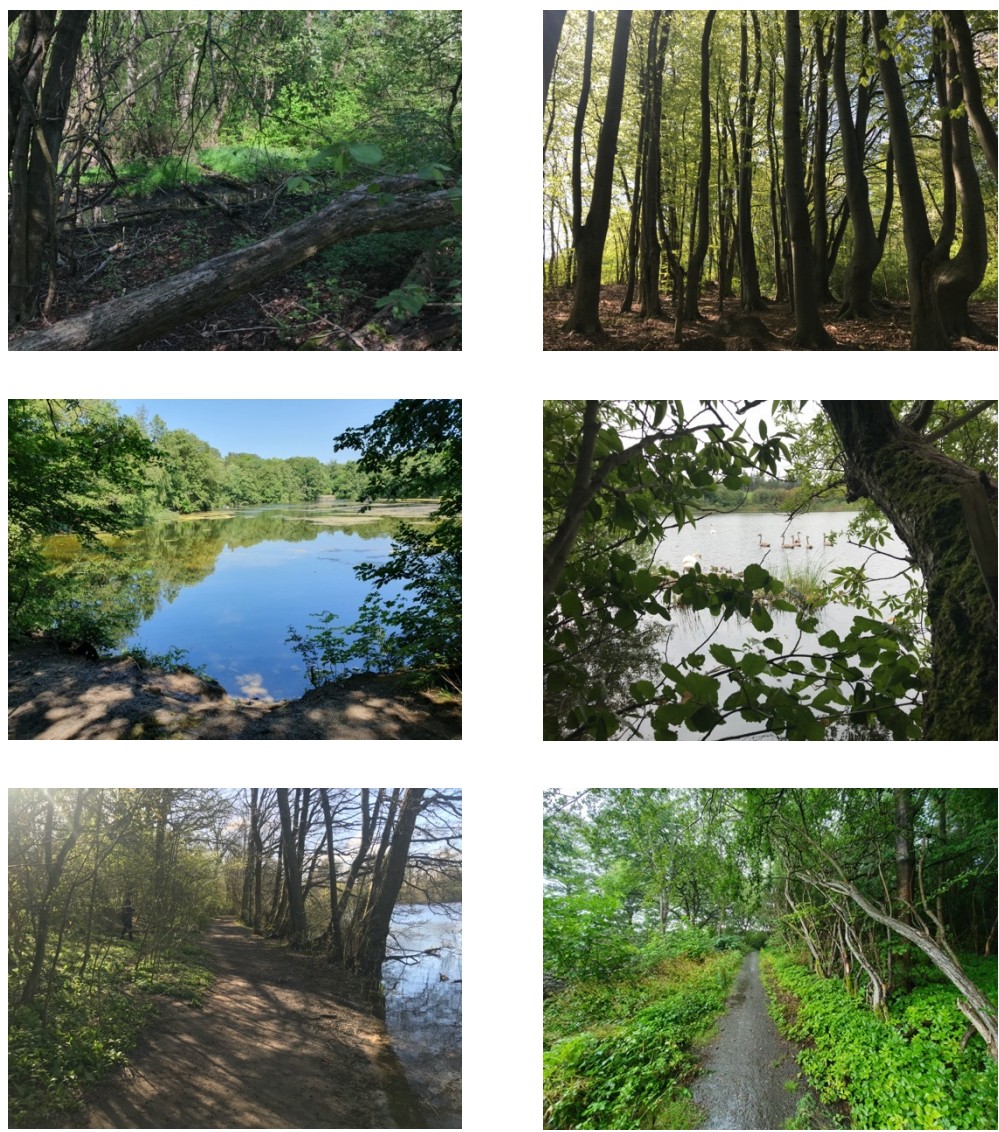

**Figure 4.** Photographs from the area that will become the Move Green Nature Trail, illustrating its rich variation of vegetation, the views over one of the two lakes, and parts of the existing trail system.

### 2.3. The Move Green Nature Trail and the First Step of the EBHDL Process Model

The first step of the EBHDL model, 'Evidence collection' started in 2015. Initially, a screening of research-based and practice-based knowledge from other Danish research environments was conducted. The screening revealed that both research and practice divulged limited evidence on people with mobility disabilities and their use of nature. Therefore, the evidence collection step of the EBHDL process model became comprehensive [33]. Evidence was collected from a wide range of sources: 1. a systematic literature review to evaluate the scientific evidence for health benefits of the design and use of green spaces for people with mobility disabilities [8]; 2. literature on the relationship between nature and human health (e.g., theoretical works); 3. statistical analyses of a nationwide survey to provide knowledge on the use of green space by people with mobility disabilities, on associations between the frequency of use and distance to nearest green space, as well as on health-related quality of life status [11]; 4. interviews with professionals that work in nature with people with mobility disabilities, to obtain information on how they work with the target group in nature and what barriers they experience [30]; 5. focus group interviews (indoors) with people with mobility disabilities, to explore the experiences, preferences, and constraints they meet when using green spaces [34]; 6. individual in-depth

interviews (out in nature) with people with mobility disabilities with the same aim as above [34]; 7. various landscape analyses of the site were conducted to identify the level of accessibility and existing health-promoting qualities (e.g., registration of spatial conditions, aesthetic assessment of natural scenery, and experiences in relation to possible restorative qualities and accessibility assessment); 8. field studies and landscape analyses of accessible green spaces in Denmark and abroad, to identify and be inspired by existing accessibility solutions [30]; 9. design seminars with practitioners and professionals working within landscape architecture, construction, accessibility, and rehabilitation, to obtain practical knowledge from professionals with experience in working with accessibility and with people with mobility disabilities in nature; 10. workshops regarding: a. physiotherapists and manual wheelchair users' experiences of rehabilitation in nature, b. mental restitution and social inclusion with users in electric wheelchairs in nature, c. accessibility and nature experiences with users in electric wheelchairs, to obtain first-hand information on the site; 11. dialogue with the authorities regarding nature protection, rules and legislation in relation to the site for the Move Green Nature Trail [35–38]; 12. dialogue with the contracting architectural company regarding technical design solutions. Much of the evidence has been published in articles [8,11,34] and in a working report [30], but some of the evidence based on the landscape analyses, interviews, workshops, and seminars are in unpublished research notes.

The collected evidence was sorted into the four topics of evidence: 'target group', 'nature and human health relationship', 'environment', and 'use of nature' (Table A1). Some evidence collection methods resulted in evidence that could be allocated to just one topic, while other methods led to evidence that could be allocated to several topics. For example, the evidence from the focus group and in-depth interviews with the target group were allocated to four topics of evidence, while evidence from the landscape analyses was only allocated to the topic 'environment'.

**Table 1.** The table illustrates the 12 ways evidence was collected in the Move Green project. The X marks which of the four topics of evidence (from step 2 of the EBHDL process model) are related to the evidence collected.

| | How the Evidence Was Collected | Target Group | Nature and Human Health Relationship | Environment | Use of Nature |
|---|---|---|---|---|---|
| 1 | Systematic literature review evaluating the scientific evidence for health benefits of the design and use of green spaces for people with mobility disabilities | X | | X | X |
| 2 | Literature on the relationship between nature and human health (e.g., theoretical works) | | X | | |
| 3 | Statistical analyses of nationally representative health data | X | | X | |
| 4 | Interviews with professionals working in nature with people with mobility disabilities | X | | X | X |
| 5 | Focus group interviews (indoors) with people with mobility disabilities | X | | X | X |
| 6 | Individual in-depth interviews (in nature) with people with mobility disabilities | X | | X | X |

**Table A1.** *Cont.*

| | How the Evidence Was Collected | Target Group | Nature and Human Health Relationship | Environment | Use of Nature |
|---|---|---|---|---|---|
| 7 | Landscape analyses | | | X | |
| 8 | Field study of accessible green spaces in Denmark and abroad | | | X | |
| 9 | Design seminars with practitioners and professionals working with landscape architecture, construction, accessibility and rehabilitation | | | X | X |
| 10 | Workshops regarding -Rehabilitation with physiotherapists and users in manual wheelchairs in nature -Mental restitution and social inclusion in nature with users in electrical wheelchairs -Accessibility and nature experiences with users in electrical wheelchairs | X | | X | X |
| 11 | Dialogue with authorities regarding nature protection and rules within the chosen area for the Move Green project | | | X | |
| 12 | Dialogue with the contracting architectural company regarding the design. | | | X | |

## 2.4. Working Method for the Programming of the Move Green Nature Trail

According to the EBHDL process model, the programming should start by formulating the aim of the design [20]. The aim of the design for the Move Green Nature Trail was predetermined by the overall goal of the Move Green Research Project: to help people with mobility disabilities improve their mental, physical and social health through mental restoration, physical exercise, and social activities. The process of identifying and systemizing the evidence in the programming was not linear. Much evidence had been identified before the programming step (1–10, in Table A1), but new evidence was also gathered later in the process (11–12 in Table A1). As the evidence was collected, it was added to a gross list of evidence points in a table, organized under the four main topics. The list was then scrutinized to avoid double entries, and a final decision was made about which topic headings the evidence points were to be placed under. The decision on which of the evidence points to include in the final program was made by the responsible landscape architects of the project, based on their professional expertise and competencies. The description of the evidence points was formulated so that people with various backgrounds and varying levels of knowledge of the field of research and practice could understand the content. The evidence points were then translated into specific design criteria. Design criteria are explicit goals that must be achieved by the design in order to for the design to be deemed successful. Based on the design criteria, design solutions were generated and described. These were site-specific practical solutions on how each of the design criteria were to be solved by the design.

## 3. Results

### *3.1. Result of the Programming: The Program*

The aim of the design for the Move Green Nature Trail is to offer people with mobility disabilities improved mental, physical, and social health through mental restoration, physical exercise, and social activities. To fulfil this design aim, a program was developed (see Appendix A, Tables A1–A4). As well as being a design tool for the Move Green Nature Trail, the program tables also lend themselves as a reporting format for the results of the programming step of the Move Green Nature Trail project presented in the present paper, which we do under the four topics 'Target group', 'Nature and human health relationship', 'Environment', and 'Use of nature'.

#### 3.1.1. Target Group

Six points of evidence were selected for the topic 'Target group' (see Appendix A, Table A1). The evidence was mainly focused on who the design was aimed at, and concerned issues such as health, disability, use of assistive devices, and personal barriers to visiting nature. The evidence was extracted from the systematic literature review, the statistical analyses of the nationally representative health data, the focus group interviews, the individual in-depth interviews, and from the working report presenting the interviews with professionals and notes from workshops (see Appendix A, Table A1 for the specific references). The evidence covered many disciplines such as public health, landscape architecture, and physiotherapy. Based on the selected evidence, eight design criteria and nine design solutions were formulated. Examples of design solutions with a more overall character included dividing the area into three trail systems that would offer different challenges, experiences and activities, and integrating the design into the existing landscape. Practical design solutions were also formulated, for example for the dimensions of the trails.

An example of an evidence point for the topic 'Target group' is: "wheelchair users have eye level at approx. 120 cm above terrain" (Appendix A, Table A1). The evidence stemmed from the national guide on accessible outdoor environments [35]. Since an individual's eye height varies depending on their ability to walk, their height, and their use and type of assistive devices, the design criteria were set as: "The design must take different heights of eye level into consideration". The design solution was: "Different eye heights are accounted for in relation to views, experiences, planting and constructions". This design solution was of general character and was to be applied throughout the site.

#### 3.1.2. Nature and Human Health Relationship

Six points of evidence were selected for the topic 'Nature and human health relationship' (see Appendix A, Table A2) and focused on the relationship between nature and human health and the intended health outcomes. The evidence for this topic was obtained from white literature documenting positive relations between nature and mental, physical, and social health, and theories explaining these relationships (see Appendix A, Table A2 for the specific references). The theories mostly stem from the disciplines of environmental psychology, human geography, and landscape architecture.

Six design criteria and 18 design solutions were formulated based on the evidence. While the 18 design solutions accommodated many details, their essence was about how the trails and the surrounding landscape held qualities for restorative environments and experiences of protection and safety. Further, the design solutions dealt with the possibilities for mental, physical, and social activities in relation to the three trails.

An example of an evidence point for the topic 'Nature and human health relationship' is: "Prospect: Open views over long distances and brightly lit. Refuge: Place of concealment and protection, small, and dark. Prospect and refuge are opposite perceptions and must occur at the same time" (Appendix A, Table A2). The evidence point stemmed from the Prospect-Refuge theory of Jay Appleton [39]. This theory relates to satisfying inherent desires of opportunities (prospect) and safety (refuge) in the landscape. The following

design criterion was formulated: "The design should offer prospect/refuge experiences". Consequently, the design solution became: "Experiences of refuge (terrain and vegetation) with prospects over bright areas (ponds and meadows) are offered along the trails". The ambition of the design solution was to create safe refuge areas with the help of terrain and vegetation strategically placed with visual access over the lakes and open fields.

### 3.1.3. Environment

Six points of evidence were selected for the topic 'Environment' (see Appendix A, Table A3). The environment itself and its relation to the characteristics of the target group are the main foci. The evidence was obtained from the systematic literature review, the statistical analyses of the nationally representative health data, the focus group interviews, the individual in-depth interviews, the interview with professionals, the landscape analyses, the field studies, the design seminars, the workshops, and the dialogue with authorities and with the contracting architectural company (see Appendix A, Table A3 for the specific references). This evidence was found within the fields of landscape architecture, health science, planning, and nature management.

Sixteen design criteria and 18 design solutions were formulated based on the evidence. The design solutions focused on how the landscape should direct the course of the trails, and stipulated that all trails were to be kept level and well maintained. The existing vegetation was to be preserved and enhanced, thus keeping the natural and wild appearance of the site. The design solutions also stipulated that visitors were to be able to have physical contact with nature, both vegetation and terrain, on the trails and on off-trail paths. General facilities such as handicap-accessible toilet facilities and parking were to be provided. The area is protected by legislation and regulations, and building permissions were to be applied for at the municipality.

An example of an evidence point for the topic 'Environment' is: "Trails must be passable for people using assistive devices. Avoid physical barriers" (Appendix A, Table A3). The evidence point stemmed from articles [8,34], a working report [30], the national guide on accessible outdoor environments [35], and notes from landscape analyses, field studies, and workshops. Physical barriers hinder people with mobility disabilities from visiting green spaces, thus the design criteria focused on how to overcome this: "The trails must be accessible along their entire length. Avoid loose and uneven materials (e.g., sand, loose gravel, pebbles, grass reinforcement). In the design, physical barriers (e.g., steps, steep trails, cattle grids and gates) must be replaced with integrated and accessible natural-looking design solutions". Further, the design criteria specified that the design solutions should not be "add-ons" but rather integrated into the landscape. The following design solutions were formulated: "Smooth, firm, barrier-free ground surface. Steps are replaced with modulated terrain. To integrate the trails into the landscape, natural ground surface to be used to the largest extent possible and with respect for existing plants. Where it is not possible to modulate the natural surfaces, raised trails are to be constructed". The ambition of these design solutions was to make the site accessible with respect for and with inspiration from the existing landscape and vegetation.

### 3.1.4. Use of Nature

Seven points of evidence were selected for the topic 'Use of nature' (see Appendix A, Table A4). The focus for this topic was what characteristics the target group has and what the environment should offer them. The evidence was obtained from the systematic literature review, the interviews with professionals, the focus group interviews, the individual in-depth interviews, the design seminars and the workshops, and was primarily found within the discipline of landscape architecture (see Appendix A, Table A4 for the specific references).

Ten design criteria and 20 design solutions were formulated based on the evidence. The design solutions focused on information about the area, but also on what the design should offer in relation to experiences and activities in supporting mental health (possibility

for awareness exercises in serene, calm environments with views over water), physical health (possibility for physical training of various intensity, determined by the length of the trails, slopes etc.), and social health (social activities in and with nature). Sensory experiences and stimulation were given by access to water and resting possibilities.

An example of an evidence point for the topic 'Use of nature' is: "Access to and view over water seems soothing and gives sensory stimulation for PwMD" (Appendix A, Table A4). This evidence stemmed from two articles [8,34] and notes from workshops. People with mobility disabilities are often restricted to only being able to view water from a distance. However, enabling sensory stimulation from water is also desirable. Therefore, the following design criteria were formulated: "The design should offer the possibility to view water, to get close to water and to get out on and in water". The following design solutions were formulated: "All trails offer views over water areas. The course of the trail is selected to offer different kinds of water (lakes, stream, swamp). Parts of the trail stretch out into the water. Bridges and boardwalks make it possible to cross water. A floating bridge gives the sensation of water movement. A platform offers physical contact with water. Where needed, railings and safety borders are present". In addition to offering views, the design solutions offer themselves to more sensory stimulation with the opportunity to get on water and in water.

## 4. Discussion

This paper has presented an implementation of the four-step EBHDL process model, focusing on the programming step of the model, as an example of how landscape architects can work with an evidence-based approach to support human health through accessible green spaces. The model can be seen as a response to the concerns regarding the importance of ensuring that professionals have the skills, tools, and knowledge to design green space that delivers the desired positive health outcomes and thereby enhances sustainability.

The paper has demonstrated how the programming step of the EBHDL process model was implemented in the designing of the Move Green Nature Trail. The aim of the design as formulated in the program presented in the current paper was to design a nature trail in the Hoersholm arboretum that is intended to offer people with mobility disabilities improved health. The programming phase resulted in 25 points of evidence divided into the four topics: 'Target group', 'Nature and human health relationship', 'Use of nature', and 'Environment'. In turn, the evidence was translated into 37 design criteria and 65 design solutions. The programming was conducted by three of the authors who are also landscape architects. This context explains why the level of detailing is relatively high in the program for the Move Green Nature Trail, which ensures that it is possible to identify where the evidence points came from and how they have been processed. If the EBHDL process model was to be used at a practicing landscape architect company, it may also be useful and appropriate that they use the same level of detail in their program, since many people may be involved in the design process and may be so at different stages, and would therefore need access to the details and thinking behind the program. However, the EBHDL process model does not stipulate how detailed a program should be, rather the level of detail should be adapted to the specific case the model is to be applied to. In the current study, attention was paid to ensuring that the level of detail in the program should allow anyone to identify the grounds of the decisions made by the landscape architects in their programming. This transparency in the programming step is essential for the fourth step in the process model—'Evaluation'. This should be considered by users of the EBHDL process model, as it allows designers to revise a program and identify design criteria and solutions that work as they were intended to.

During our programming, it was challenging to categorize the evidence points under the four main topics of the EBHDL process model 'Target group', 'Nature and human health relationship', 'Environment' and 'Use of nature'. The topics should be understood as areas from which it is relevant to collect evidence. Often the topics are interrelated, and therefore the division of them into the four topics is not always obvious. A key factor is

that all relevant points of evidence must be included in the program. In relation to the program presented in the current paper, the placement of some of the evidence points in relation to the four topics can be debated. For example, the point of evidence "Lack of general information about accessibility and wayfinding in the area prevents visits from PwMD" has been placed under the topic 'Use of nature'. The reason for this is that it relates the target group's possibilities to use the green space. One could argue that the point of evidence could also be placed under the topic 'Environment', since it relates to the existing conditions. However, the placement of such a point does not influence the final design solutions, as solutions based on the evidence point will be designed as long as it is still included in one of the topics under programming.

In relation to how to design health-promoting green space, design guidelines are often presented in the literature, which can be included as evidence in an EBHDL process model. Such design guidelines often comprise sets of specific design recommendations for landscape architects [27]. Often, specific guidelines are developed for certain patient groups, for example people suffering from dementia or people suffering from stress-related illnesses. However, such guidelines can seldom stand alone, as no projects are completely alike, since the project developers' ambitions, the intended use, the existing site conditions, and budget may vary. The structure and systematics of the program of the EBHDL process model mean that it can be applied in any health design project. In projects that are alike, the parts of the evidence collected will be the same, for example the same target group, same aim, and plans of use. However, evidence regarding environment will most certainly be different. The design criteria could have some similarities, but there will be great differences regarding design solutions. Therefore, programs developed for different projects do not look the same. There is not one single program that can totally be transferred and used in other design projects, each project and each program is unique.

The EBHDL process model uses a broad evidence definition, including evidence from different fields of research, but also from practice, which allows for the inclusion of evidence from sources that are highly relevant for and familiar to practising landscape architects, for example landscape analyses, workshops, best practices, and design guidelines. It can be argued that this broad perspective on evidence is not in line with the common evidence-based design approach, for example as defined by Brown and Corry [21]. The broad approach to collecting evidence in the EBHDL process model attempts to address the current gap between research and practice in the field of health design and landscape architecture [15]. This gap can be partly attributed to a lack of focus on the role of landscape architecture in research on nature and human health relationship, thus little research-based evidence is available for practising landscape architects to apply in their design work [15].

Though the quality of some items in the broad range of evidence that can be included in the EBHDL process model was not directly assessed in the selection of evidence for the Move Green Nature Trail program, the final step in the model, the post-occupancy evaluation, was designed to reveal if the evidence included in the program was adequate and if the landscape architect has interpreted it and translated it appropriately. The EBHDL process model should therefore be understood as a guide for the landscape architect, not a 'recipe'. Merely working in an evidence-based manner does not automatically mean that a design will have a health-promoting effect.

The EBHDL process model requires the landscape architect to be able to move between objectivity and subjectivity without being biased. Bias may occur if the landscape architect excludes evidence points that do not fit his or her personal preferences, taste and likes. In formulating an EBHDL program, objective evidence gathered from peer-reviewed research articles must be selected subjectively, based on what the landscape architect believes will be beneficial for the design. Possible workshops and landscape analyses should be conducted as objectively as possible, such that all participants' views and inputs are noted, or all potential landscape resources are identified. However, this open information gathering must be balanced with the overall goals of a specific project. The EBHDL process model requires that the landscape architect subjectively formulates the design criteria and design

solutions. One could say that along with the programming (evidence, design criteria, and design solution), the model requires an increasing gradient of working more subjectively as the landscape architect starts to work creatively towards the final design.

## 5. Limitations and Future Perspectives

The most significant drawback in choosing to apply the EBHDL process model is that it is time consuming and thereby costly. The budget for a project that uses the EBHDL process model must also include the costs of conducting the post-occupancy evaluation and any subsequent design alternations. This might prove to be a hurdle for some projects. The Move Green Nature Trail project was preceded by a pilot project where much of the evidence was collected [30]. The pilot project was initiated because people with mobility disabilities were a new target group for us, and we suspected that not much research-based evidence existed on this target group. During the pilot project we conducted several research studies, and thereby obtained much of the evidence we used in the programming of the Move Green Nature Trail project. It is of course not realistic to expect a practicing landscape architectural company to carry out such comprehensive pilot projects and research themselves.

Though the Move Green Nature Trail project focusses on a heterogenous (e.g., broad variations in functioning, use of different assistive devices, large age span) target group of people with mobility disabilities, the project has not accounted for groups with other types of disabilities. This may be regarded as a limitation of the project. Further, accessibility in landscape architecture often refers to a minimum of barriers for people with disabilities. While universal design is often mentioned in relation to accessibility, universal design should not be understood as a synonym for accessibility. Universal design is a concept that has developed over time, and today it can be understood as many things and can be applied in different ways [40]. In the CRPD universal design is defined as: "the design of products, environments, programs and services to be usable by all people, to the greatest extent possible, without the need for adaptation or specialized design" [41], which many practitioners find overwhelming and unmanageable to use in their practice. From a practical perspective universal design is often viewed as a set of principles that could guide the design process in order to avoid exclusion or be used as an evaluation tool [42]. More recently, universal design has been viewed more holistically and has been defined as "a process that enables and empowers a diverse population by improving human performance, health and wellness and social participation" [43]. There are many similarities between this definition and the EBHDL process model, and it would be beneficial in future projects to inscribe the concept of universal design into the EBHDL process model.

The Move Green Nature Trail is to be constructed in 2021. Thereafter the fourth step (evaluation) of the EBHDL process model will start. This will allow us to assess if the design works as intended, or if more evidence and re-design are needed. The EBHDL process model has also been used in the design of Nacadia® therapy garden [44] and Octovia® health forest [45]. Even though the results from these projects are promising [31,32] and can be seen as the first steps in the process of validating the EBHDL process model, the model needs to be used in many more projects in order to be validated.

To design in an evidence-based manner, relevant and applicable evidence must be available to the designer. In projects for specific target groups, it can be challenging to find evidence. We recommend that more landscape architects turn their hand to research and contribute with study designs for various target groups that will provide evidence that can be applied in related future designs. Furthermore, we encourage landscape architectural companies to hire landscape architects that are trained in research and evidence-based design.

In some parts of the world, we see that the authorities require designers to be qualified in evidence-based design, for example in Australia, Singapore, and the United Kingdom designers are required to be qualified in evidence-based design if they are to design hospitals [27]. While this is not the case in Scandinavia, we encourage teachers of landscape

architecture to teach their students how to work in an evidence-based manner. Hopefully, in the near future, a new generation of landscape architects will enter the labor market with skills in evidence-based health design in landscape architecture.

## 6. Conclusions

The WHO has stated that it is vital that green space is designed in a manner that delivers positive health outcomes. Evidence-based design processes are recommended as a means of ensuring such positive health outcomes. One fundamental problem in the field of health design in landscape architecture is that many landscape architects are not trained in evidence-based design, and therefore lack the required knowledge and skills. Leading scholars in evidence-based design in landscape architecture have therefore called for methods or guiding questions to help landscape architects work in an evidence-based manner. In this paper, we have applied the EBHDL process model for evidence-based design in landscape architecture as a response to that call. This comprehensive model includes the whole design process, divided into four steps: 'evidence collection', 'programming', 'designing', and 'evaluation'. This paper has described the implementation of the second step of the model, programming, on Move Green Nature Trail project. The programming step systematizes the collected evidence, and consists of three parts: a. 'Aim(s) of the design', i.e., what health outcomes should the design have an impact on; b. 'Design criteria', which are explicit goals based on the evidence; and c. 'Design solution', which describes how the design criteria will be solved in the design. It is our hope that this paper demonstrates how evidence can be translated into clear design solutions that form the basis of a design that meets the needs of a target group. To a large extent, people with mobility disabilities are today left behind in relation to visiting green space, and are thus excluded from the potential health benefits of interaction with green space. Designing health-promoting and accessible green space contributes to the sustainable development agenda, the SDGs, and the pledge to 'Leave no one behind', but leaves the landscape architect with a great responsibility and challenge to solve. The EBHDL process model may contribute to helping landscape architects to solve this difficult but important challenge.

**Author Contributions:** Conceptualization, M.C.G. and U.K.S.; Formal analysis, M.C.G.; Funding acquisition, U.K.S.; Investigation, M.C.G. and G.Z.; Methodology, M.C.G., U.S., and U.K.S.; Project administration, M.C.G. and U.K.S.; Supervision, U.K.S.; Visualization, M.C.G. and U.S.; Writing—original draft, M.C.G. and U.K.S.; Writing—review & editing, M.C.G., U.S., G.Z., and U.K.S. All authors have read and agreed to the published version of the manuscript.

**Funding:** This research was funded by Bevica fonden (ID436), Lokale og Anlægsfonden (2018-F-0001), Nordea-fonden (02-2018-1391) and 15. Juni Fonden (2018-H-31A and 2108-H-31B).

**Institutional Review Board Statement:** Not Applicable.

**Informed Consent Statement:** Not Applicable.

**Data Availability Statement:** Not Applicable.

**Acknowledgments:** We extend a special thanks to all the people who participated in the workshops.

**Conflicts of Interest:** The authors declare no conflict of interests.

# Appendix A

**Table A1.** The table represents the programme for the Move Green Nature Trail regarding the topic 'Target group'.

| Programme for the Move Green Nature Trail<br>Target Group | | |
| --- | --- | --- |
| Aim of the design: Nature trail offering people with mobility disabilities (PwMD) improved mental, physical and social health through mental restoration, physical exercises and social activities. | | |
| **Evidence** | **Design Criteria** | **Design Solution** |
| Gender, age, severity of disability and use of assistive devices may affect the perception of the nature environment.<br><br>Evidence derived from:<br>Ref. [8,34] and notes from workshops | The design should offer nature experiences, activities and challenges corresponding to the target group. | The area is divided into three trail systems (green, red, black) offering different experiences, mental and physical challenges and activities. Nodes connect the trail systems, making it possible to change trail along the way. |
| Accessible design solutions that are not integrated into the landscape (add-ons) expose the users and may make them feel observed.<br><br>Evidence derived from:<br>Ref. [34] and notes from workshops | Avoid distinct design solutions for PwMD. | Integrate accessibility solutions so that the setting can be used on equal terms by all users. |
| PwMD may have an assistant person with them.<br><br>Evidence derived from:<br>Ref. [8,34] | The design should offer enough space for both persons. | Adequate turning and passage spaces on trails. |
| PwMD are generally challenged with poorer mental, physical and social health, and lower health-related quality of life compared to the able-bodied population. Health-related quality of life is related to frequency of use of nature.<br><br>Evidence derived from:<br>Ref. [11] | The design should have a positive impact on PwMD's mental, physical and social health and health-related quality of life. The design should offer nature experiences (perceived sensory dimensions and the four components from attention-restoration theory that constitute restorative environments) and activities that are preferred by the target group. | Nature experiences offering mental restoration.<br>Possibilities for different levels of physical training in and with nature along the trails and in specific spaces. Possibilities for both social and solitary activities. |
| Wheelchair users have eye level at app. 120 cm above terrain.<br><br>Evidence derived from:<br>Ref. [35] | The design must take different heights of eye level into consideration. | Different eye heights are accounted for in relation to views, experiences, planting and constructions. |
| PwMD experience intra- and interpersonal barriers when visiting nature.<br><br>Evidence derived from:<br>Ref. [8,30,34] | Everyone should be able to use the trails on equal terms.<br>The design must offer a safe walk. | Integrated solutions that are accessible and safe to everyone. |

**Table A2.** The table represents the programme for the Move Green Nature Trail regarding the topic 'Nature and human health relationship'.

| Programme for the Move Green Nature Trail Nature and Human Health Relationship | | |
|---|---|---|
| Aim of the design: Nature trail offering people with mobility disabilities (PwMD) improved mental, physical and social health through mental restoration, physical exercise and social activities. | | |
| **Evidence** | **Design Criteria** | **Design Solution** |
| Green spaces have a positive impact on human health in three ways: Providing mental (and psychological) restoration; Encouraging physical activity; Encouraging social contact Evidence derived from: Ref. [2,46–50] | The design should offer possibilities for mental restoration, physical activity and social contact. | Possibilities for mental restoration and guided experiences in nature along the trails and in specific spaces. Possibilities for physical training in and with nature along the trails and in specific spaces. Possibilities for both social and solitary activities. |
| Attention restoration theory: Humans have two types of attention: 'directed attention' and 'undirected attention'. Directed attention is used when the individual has to concentrate on important matters, which requires effort and can cause mental fatigue. While stimuli that captures the individual's attention effortlessly (the undirected attention) can lead to recovery from mental fatigue Evidence derived from: Ref. [51] | In order for an environment to be perceived as restorative it should offer the following components: Being away Extent Fascination Compatibility | All four components should be present throughout the area. Mental and physical distance to everyday environment. The terrain, vegetation and water features offer a rich and coherent environment providing the users with a feeling of 'extent'. The area provides interesting nature experience to engage the mind. Information about the levels of accessibility along the trails and general information is given, e.g., distance to benches, length of trails, parking, restrooms, maintenance. |
| *Prospect-Refuge theory:* Prospect: Open views over long distances and brightly lit. Refuge: Place of concealment and protection, small and dark. Prospect and refuge are opposite perceptions and must occur at the same time. Evidence derived from: Ref. [39] | The design should offer prospect/refuge experiences. | Experiences of refuge (terrain and vegetation) with prospects over bright areas (ponds and meadows) are offered along the trails. |
| *Supportive environment theory pyramid:* Users' experience of nature and the level of demands they are able to cope with depend on their emotional and cognitive resources. This is illustrated as a four-level pyramid where the need for green spaces with less demands is large at the bottom and minor at the top level. Evidence derived from: Ref. [52] | The design should address all four levels of the pyramid in relation to mental, physical and social demands. | All four levels are present in all three trails. The locations representing the different levels are placed such that they avoid conflicts between highly demanding and less demanding environments. The area offers off-track trails with possibilities to come even closer to nature on one's own or with others. |

**Table A2.** *Cont.*

| Programme for the Move Green Nature Trail Nature and Human Health Relationship | | |
| --- | --- | --- |
| *Affordance theory:* Affordances are functional properties of an environmental feature relative to an individual that indicate what one can do in the setting and what activities may be ruled out.<br><br>Evidence derived from: Ref. [53,54] | The design should motivate meaningful functions that are possible for the individual user to do. | Environmental features offering multiple meaningful functions are distributed along the trails. The user groups' diversity (age, mobility disability, use of assistive devices, experiences with being in nature etc.) is accounted for. In line with the three different trails, there is a progression in the coding of the affordances, meaning the black trail is less coded and offers more multifunctional activities. |
| *Perceived sensory dimension (PSD):* People perceive green space in terms of certain dimensions, some more important and preferred than others. People in general prefer the dimension 'serene', followed by 'space', 'nature', 'rich in species', 'refuge', 'culture', 'prospect' and 'social'. The dimensions 'refuge' and 'nature' are most strongly correlated with stress. A combination of 'refuge', 'nature' and 'rich in species', and a low or no presence of 'social', could be interpreted as the most restorative environment for stressed individuals.<br><br>Evidence derived from: Ref. [55] | All eight PSDs should be represented in the design | Parts of the trails target mental health: 'refuge', 'nature' and 'rich in species'. Parts of trails target physical health: 'prospect', 'refuge', 'rich in species' and 'space'. Parts of the trails target social health: 'social' (dominating) in combination with other PSDs. |

**Table A3.** The table represents the programme for the Move Green Nature Trail regarding the topic 'Environment'.

| Programme for the Move Green Nature Trail Environment | | |
|---|---|---|
| Aim of the design: Nature trail offering people with mobility disabilities (PwMD) improved mental, physical and social health through mental restoration, physical exercises and social activities. | | |
| **Evidence** | **Design Criteria** | **Design Solution** |
| PwMD are excluded from experiencing wild/untouched nature (for example variation in spaciousness and type of vegetation), but want the same nature experiences as the able-bodied population, e.g., to get physically close to and able to sense nature.<br><br>Evidence derived from:<br>Ref. [11,30,34], notes from workshops, field studies and design seminars | Existing conditions (location of existing trees, stones, terrain etc.) on the site should guide the design of the course of the trail.<br>Preserve and enhance the nature experiences and nature/wildlife.<br>The design should strengthen the wild natural qualities so that the users are able to get close to varied terrain, vegetation and water.<br>Accessibility solutions should be integrated into the landscape and be used by all.<br>The nature qualities and experiences should not be destroyed when making the site accessible. | The existing landscape steers the trails' direction and curves. Different solutions should be adapted to the terrain with respect to the existing conditions.<br>Plants need to grow 'wild' and in their natural form - preferably in several layers, so that the user can get close to them irrespective of the person's height.<br>Choose 'natural' materials and colours that blend into the environment.<br>Keep existing and new vegetation close to trails (on the sides and above).<br>Offer the possibility of leaving the main trails by making off-track trails.<br>The existing wildlife and plants are respected by using plants from earlier plant list to ensure consistency between new and old planting. |
| Trails must be passable for people using assistive devices. Avoid physical barriers.<br><br>Evidence derived from:<br>Ref. [8,30,34,35], notes from landscape analyses, field studies and workshops | The trails must be accessible along their entire length.<br>Avoid loose and uneven materials (e.g., sand, loose gravel, pebbles, grass reinforcement).<br>In the design, physical barriers (e.g., steps, steep trails, cattle grids and gates) must be replaced with integrated and accessible natural-looking design solutions. | Smooth, firm, barrier-free ground surface. Steps are replaced with modulated terrain.<br>To integrate the trails into the landscape, natural ground surface to be used to the largest extent possible and with respect for existing plants.<br>Where it is not possible to modulate the natural surfaces, raised trails are to be constructed. |
| A high level of safety is needed to prevent accidents.<br><br>Evidence derived from:<br>Ref. [8,34,35] and notes from workshops | It must be safe and secure to travel along all three trails.<br>A certain maintenance level is required to ensure safety, but the design must have as low a maintenance level as possible due to costs. | Where needed, railings and safety edges are provided.The maintenance level must be equal to other similar green spaces, meaning no clearing of leaf fall or of snow, but physical barriers (e.g., fallen tree branches on the trail) are to be removed and safety check-ups are to be done regularly.<br>Inform users about the level of maintenance according to seasons.<br>Information on trails is to be provided online and on site about expected seasonal disturbances. |
| Lack of (handicap) toilet facilities can prevent PwMD from visiting green spaces.<br><br>Evidence derived from:<br>Ref. [30,34,35] and notes from field studies | There must be access to minimum one handicap-accessible toilet (from both sides). | Visitors can use existing accessible toilet located close to the entrance.<br>Access to existing facilities is barrier free. |

<div align="center">**Table A3.** *Cont*.</div>

| Programme for the Move Green Nature Trail Environment | | |
|---|---|---|
| Lack of (handicap) parking facilities close to entrance can prevent PwMD from visiting green spaces.<br><br>Evidence derived from:<br>Ref. [30,34,35] and notes from field studies | Minimum one handicap parking space. Parking must be a maximum of 150 meters from the entrance. | Parking is located adjacent to the entrance – two handicap parking lots (van) are located closest to the entrance. Both lots are marked on the ground and with raised handicap signs. |
| Legislation, protection notices and regulations:<br>Municipal and district plans<br>Section 3 protected areas<br>Danish Standard (recommendations)<br>Building Regulations (BR18)<br><br>Evidence derived from:<br>Ref. [35–38] and notes from meetings with authorities and contracting architectural company | Necessary permits must be obtained from the municipality.<br>The trails should function as a 1:1 test site and all relevant recommendations will be considered, but not necessarily fulfilled.<br>Information about the project will be presented to relevant stakeholders before construction. | Depending on the municipality's decision, possible design corrections may be made. |

**Table A4.** The table represents the programme for the Move Green Nature Trail regarding the topic 'Use of nature'.

| Programme for the Move Green Nature Trail Use of Nature | | |
|---|---|---|
| Aim of the design: Nature trail offering people with mobility disabilities (PwMD) improved mental, physical and social health through mental restoration, physical exercises and social activities. | | |
| **Evidence** | **Design Criteria** | **Design Solution** |
| PwMD need resting options corresponding to their individual needs.<br><br>Evidence derived from:<br>Ref. [8,30,34] and notes from workshops | The design must offer resting opportunities along the trails including benches.<br>The benches vary in accessibility corresponding to the degree of difficulty of the three trails. | Resting options are distributed throughout the trails with the closest distance between them along the green trail.<br>The accessibility of the sitting facilities ranges from natural elements (stones or tree stumps) to highly accessible benches with backrests and armrests. |
| Access to and view over water seems soothing and gives sensory stimulation for PwMD.<br><br>Evidence derived from:<br>Ref. [8,34] and notes from workshops | The design should offer the possibility to view water, to get close to water and to get out on and in water. | All trails offer views over water areas. The course of the trail is selected to offer different kinds of water (lakes, stream, swamp). Parts of the trail stretch out into the water.<br>Bridges and boardwalks make it possible to cross water. A floating bridge gives the sensation of water movement. A platform offers physical contact with water. Where needed, railings and safety borders are present. |
| PwMD may want to visit the trails alone or in company with others (with or without disabilities).<br><br>Evidence derived from:<br>Ref. [8,34] | The trail should be designed for PwMD, but also people without disabilities should be able to walk along it. | All design solutions are integrated and can be used by people with and without mobility disabilities. |

**Table A4.** *Cont*.

| Programme for the Move Green Nature Trail Use of Nature | | |
|---|---|---|
| Lack of general information about accessibility and wayfinding in the area prevents visits from PwMD.<br><br>Evidence derived from:<br>Ref. [8,34,35] | Easy access to clear information about the area including resting options, toilet facilities, parking facilities, length of trails, slopes etc. should be availiable both online and on-site.<br>Information should be readable for people using assistive devices and integrated into the landscape. | Information board at the entrance with information on toilet facilities, parking facilities, resting options, trails (length and width, grade range, cross slope, severity), expected maintenance, fishing, biking, dogs etc.<br>Folders to bring along on trail. Possibility of downloading folder through QR-code on smartphone.<br>Wayfinding (colour markings) along the trails.<br>Information also available on website, and possibly related websites.<br>Information board with suitable material, height and angle for people using different assistive devices.<br>The information should be clear and coherent throughout the area.<br>The information board is integrated into the environment. |
| Green spaces can motivate and promote mental restoration for PwMD.<br><br>Evidence derived from:<br>Ref. [8,30,34] and notes from design seminars | Parts of the trails should be focused on mental restoration.<br>Avoid conflicts between areas for relaxing awareness exercises and areas for more outgoing activities. | The red trail is especially designated for relaxing awareness exercises.<br>The red trail offers off-track possibilities for serenity, calmness, privacy, close contact to nature and wildlife with views over water.<br>The environment sets the frame for awareness exercises. |
| Green spaces can motivate PwMD to interact socially.<br><br>Evidence derived from:<br>Ref. [8,30,34] and notes from design seminars | The design should offer interesting and fascinating nature treasures to experience with others. | The environment sets the frame for using geocaching as a narrative tool about the area. Inform about the unique plants, wildlife, the lakes and their function now and historically. |
| Green spaces can motivate and promote physical activity for PwMD.<br><br>Evidence derived from:<br>Ref. [8,30,34] and notes from design seminars | The design should offer possibilities for physical training in and with nature both alone and with others. | All three trails offer possibilities for physical training, however with varying degrees of intensity: different length of the trails, cross slope, gradients and surfaces.<br>Various activities offering different possibilities for muscle workouts.<br>The environment sets the frame for rehabilitation training programmes for the target group. |

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
