# Peer review of "From Evidence to Design Solution—On How to Handle Evidence in the Design Process of Sustainable, Accessible and Health-Promoting Landscapes"

_sustainability, doi:10.3390/su13063249_

Round 1
Reviewer 1 Report
Introduction: you must indicate why your research area is important, include study aims. The methods section is poorly written. The authors did not conduct a correct literature review; there are only 26 references. Discussions: limitation not included. Results: include statistical analysis. Include conclusions. How will your study advance knowledge on the topic?
Reviewer 2 Report
The article attempts to present a program, but the presentation lacks a research focused. The structure of the article is atypical and it only mimics a research article. The whole presentation must be redesigned in order to target the international research audience of "Sustainability". The introduction should analyze the previous approaches to find their shortcomings and create a framework justifying the need for their research, and present its goals or underlying hypotheses. The article should also develop a take-home message for the broad international research audience in terms of a "Conclusion" section. Detailed comments are provided for each section.
The abstract is poorly written. Although it starts in a promising note, revealing the importance of the issue addressed by the articles, the next parts do not provide too much information. They do not elaborate on the methodology and results in a comprehensive way and, most important, they do not provide any indication whatsoever on the contribution of their findings to the advancement of the field. The authors should remember that they submitted their contribution to Sustainability, an international RESEARCH journal, and research is done to keep science moving further.
The introductory part lacks a research focus. The review of the literature does not pinpoint any shortcomings of the previous approaches, including lacks, contradictions, or ambiguities. Moreover, it seems to rely more too little on the relevant literature (articles published in mainstream journals), and too much on political documents. Most important, there is no clear declaration of the research goals and/or underlying hypotheses, normally placed in the concluding paragraph of the introduction, and obviously no information on why their research is important with respect to the previously identified shortcomings.
The discussion miss an external validation section, comparing the findings with those of other similar studies carried out in other countries, identified in the literature.
The article is missing a concluding section, showing in a nutshell the contribution of their findings to the theoretical advancement of the field.
The reference list is immature, underdeveloped and dominated by non-citable items, such as unpublished theses, Internet, etc. This remark should be tied to the ones referring to the poor substantiation of the introduction and to the lack of an external validation of the results. Developing these sections in a manner compatible to an article published in a journal addressing an international research audience would, most likely, improve the structure of references too.
The authors are also advised to seek for the assistance of a native English speaker specialized in writing up research, because the presentation style does not match the one of a scientific article (e.g., use of the future tense in any other sections than those dealing with the future research directions). The authors should also pay attention to the Author Guidelines (e.g., the inclusion of the figure captions between parentheses, against the journal template, division of the references in different categories instead of forming a single list, format of references not consistent with the Guidelines - the journal names are not abbreviated). The division of the reference list in categories is customary for theses, and the authors are advised to read more research articles and understand the particular requirements of writing them up before submitting one; courses in writing up research, and even books on this topic, might help.
Round 2
Reviewer 1 Report
The paper has been improved however the introduction seems a dissertation introduction. Please rewrite the introduction. The Methods section is still poorly written.
Reviewer 2 Report
The authors have carried out an extensive revision, addressing most shortcomings. As a result, the article has increased research depth and addresses a broader audience. The structure of the article improved, but it is still unclear and impedes its understanding.
In more details, the introduction is meant to analyze the existing literature in order to identify the main shortcomings of the previous studies to justify the need for research and present the reseach goals in a way indicating how they address these shortcomings, and what the novel and original elements are. For this reason, the introduction must end with the research goals, currently placed somewhere in the middle. I suggest the authors reorganizing the text, and concluding the introduction with a section labeled "Research Goals and Objectives", including the way of addressing the shortcomings of the previous studies (lines 101-104). Since the "EBHDL process model" is a method, it pertains to the methods section, so lines 167-171 and Figure 2 should be moved to this section.
Also, the use of the past tense in the section "The Move Green Nature Trail and the first step of the EBHDL process model" suggestes that these are steps already taken by the authors, therefore this section belongs to the "Methods"; it is arguable if the sections "The EBHDL process model" and "The Move Green project" belong to the Methods or not. More likely, this is the correct answer, and in this case the Methods section should start with an overview showing that the methods are built up on these projects, then present the projects and show the next steps taken ("The Move Green Nature Trail and the first step of the EBHDL process model").
In any case, the article needs a clear separation between the introduction (dealing with the literature review and study goals), methods, results, and discussions. Currently the elements are mixed, making the article almost impossible to understand.
Round 3
Reviewer 1 Report
The abstract is still too vague.
Page 7 Methods: It still needs major rewriting for sense and flow. It is written like a report of the Move Green project
Figure 2 is too small.
Author Response
Comment 1: The abstract is still too vague.
Response 1: Unfortunately, it is impossible for us to revise the manuscript based on this comment as it is too unclear. We have contacted one of the guest editors who encouraged us to instead focus on their detailed and constructive comments, which were very helpful.
Comment 2: Page 7 Methods: It still needs major rewriting for sense and flow. It is written like a report of the Move Green project
Response 2: Unfortunately, it is impossible for us to revise the manuscript based on this comment as it is too unclear. We have contacted one of the guest editors who encouraged us to instead focus on their detailed and constructive comments, which were very helpful.
Comment 3: Figure 2 is too small.
Response 3: The figure has been enlarged in the manuscript. The original file is as well uploaded in a format that can be adjusted according to the journal’s layout.
Reviewer 2 Report
The authors have addressed all comments, and the structure of the article can now be understood. However, the introductory section has two sub-sections with a title and content saying the same thing in two different ways: "Aim and contributions" and "Research goals and objectives". They should be merged together under the title "Research goals and contribution to the field".
Author Response
Comments, reviewer 2:
The authors have addressed all comments, and the structure of the article can now be understood.
Comment 1: However, the introductory section has two sub-sections with a title and content saying the same thing in two different ways: "Aim and contributions" and "Research goals and objectives". They should be merged together under the title "Research goals and contribution to the field".
Response 1: This has now been revised accordingly.